# Distinguishing Incubation and Acute Disease Stages of Mild-to-Moderate COVID-19

**DOI:** 10.3390/v14020203

**Published:** 2022-01-20

**Authors:** Michael Müller, Johann Volzke, Behnam Subin, Christian Johann Schmidt, Hilte Geerdes-Fenge, Emil Christian Reisinger, Brigitte Müller-Hilke

**Affiliations:** 1Core Facility for Cell Sorting and Cell Analysis, Rostock University Medical Center, 18057 Rostock, Germany; michael.mue@t-online.de (M.M.); Christian.Schmidt3@med.uni-rostock.de (C.J.S.); 2Department of Cardiology, Rostock University Medical Center, 18057 Rostock, Germany; behnam.subin@med.uni-rostock.de; 3Department of Tropical Medicine and Infectious Diseases, Rostock University Medical Center, 18057 Rostock, Germany; hilte.geerdes-fenge@med.uni-rostock.de (H.G.-F.); Emil.Reisinger@med.uni-rostock.de (E.C.R.)

**Keywords:** SARS-CoV-2, COVID-19, disease phases, plasmablasts, cytotoxic T cells, IP-10, acute infection, antibodies

## Abstract

While numerous studies have already compared the immune responses against SARS-CoV-2 in severely and mild-to-moderately ill COVID-19 patients, longitudinal trajectories are still scarce. We therefore set out to analyze serial blood samples from mild-to-moderately ill patients in order to define the immune landscapes for differently progressed disease stages. Twenty-two COVID-19 patients were subjected to consecutive venipuncture within seven days after diagnosis or admittance to hospital. Flow cytometry was performed to analyze peripheral blood immune cell compositions and their activation as were plasma levels of cytokines and SARS-CoV-2 specific immunoglobulins. Healthy donors served as controls. Integrating the kinetics of plasmablasts and SARS-CoV-2 specific antibodies allowed for the definition of three disease stages of early COVID-19. The incubation phase was characterized by a sharp increase in pro-inflammatory monocytes and terminally differentiated cytotoxic T cells. The latter correlated significantly with elevated concentrations of IP-10. Early acute infection featured a peak in PD-1^+^ cytotoxic T cells, plasmablasts and increasing titers of virus specific antibodies. During late acute infection, immature neutrophils were enriched, whereas all other parameters returned to baseline. Our findings will help to define landmarks that are indispensable for the refinement of new anti-viral and anti-inflammatory therapeutics, and may also inform clinicians to optimize treatment and prevent fatal outcomes.

## 1. Introduction

The severe acute respiratory syndrome coronavirus type 2 (SARS-CoV-2) is a highly infectious and rapidly transmittable β-coronavirus that led to the global pandemic of coronavirus disease 2019 (COVID-19). As of summer 2021, more than 180 million people were infected, with almost 4 million casualties worldwide [1,2]. Of great concern are new virus variants that emerge periodically and exhibit augmented pandemic potentials, can cause infection of individuals vaccinated against the original strain, and re-infect patients previously recovered from COVID-19 [3,4].

COVID-19 is characterized by a diverse collection of isolated or combined symptoms ranging from mild to severe to life-threatening. While mild disease may proceed asymptomatic or show mere signs of a common cold, moderate and severe COVID-19 present with symptoms that range from fever, malaise and fatigue to neurological, dermatological, gastrointestinal and pulmonary manifestations [5]. Pneumonia may lead to an acute respiratory distress syndrome (ARDS), necessitating oxygen supplementation and mechanical ventilation [6,7]. Immense effort has been put into the pathophysiological and immunological profiling of the host reaction to the disease [8,9]. Thus, a dysregulated host immune response was identified that causes a life-threatening cytokine storm and subsequent immune paralysis, leading to multiple organ failure and death. And even though mild-to-moderate COVID-19 does not seem to be associated with a fulminant immune activation, more and more data point to long lasting infection related sequelae including persisting nausea, fatigue and loss of smell and taste in up to 30% of patients who recovered from COVID-19 [10,11,12]. Hence, even non-severe COVID-19 may pose a considerable hazard for global health and the international economy.

Although numerous studies have already addressed the differences between pathological profiles of severe and mild-to-moderate COVID-19, longitudinal trajectories within the circulating immune landscapes are still obscure [13,14]. However, a rapid identification of distinct disease stages might in the future allow for refined strategies with immunomodulatory therapies that support the host’s immune response. These therapies may indeed gain importance as more and more variants may emerge that—despite efficient vaccination strategies—require the containment of viral loads after infection.

In order to rule out any bias from immunosuppressive therapy in severe COVID-19, we chose to concentrate on mild-to-moderate cases and to analyze the peripheral blood immune cell composition and its activation, plasma levels of cytokines, and SARS-CoV-2 specific immunoglobulins. We then examined the complementary data sets for transient alterations in consecutive samples, indicative for differently progressed disease. Finally, we suggested distinct disease phases and characterized their immunological and serological layout.

## 2. Materials and Methods

### 2.1. Study Population

COVID-19 patients were recruited either from the local COVID-19 test center or from the division of tropical medicine and infectious diseases. After an initial withdrawal of peripheral blood on the day of testing or admittance to the hospital (day one), subsequent venipunctures followed on days three and seven. Individuals recruited at the test center who were tested negative were considered healthy controls and they were required to provide blood only once on the day of testing. Apart from sex and age, no other clinical parameters like co-morbidities or underlying conditions were documented for COVID-19 patients and healthy controls, respectively. The study was approved by the ethics committee of the University Medical Center of Rostock. It is filed under A 2020-0086 and written informed consent was obtained from all participants.

### 2.2. Flow Cytometry

For the analysis of surface expression markers, 100 µL of anticoagulated COVID-19 patient or healthy donor blood was used. In order to reduce unspecific antibody-conjugate binding, 10 µL fetal calf serum (Thermo Fisher, Waltham, MA, USA), 5 µL True-Stain Monocyte BlockerTM and 5 µL anti-Fc receptor TruStain FcXTM (Biolegend, San Diego, CA, USA) were added and incubated for 15 min on ice. The following amounts of antibody:fluorophore-combinations were used: 0.25 µg CD127:APC/R700 (clone HIL-7R-M21), 1 µg CD147:BV421 (TRA-1-85), 0.5 µg CD45RO:BV480 (UCHL1, BD Biosciences, Franklin Lakes, NJ, USA), 1 µg CD11b:PerCP/Cy5.5 (ICRF44), 0.8 µg CD11c:BV785 (3.9), 0.56 µg CD14:BV510 (63D3), 0.13 µg CD16:BV650 (3G8), 0.06 µg CD19:APC/Fire810 (HIB19), 0.13 CD20:SparkNIR685 (2H7), 0.5 µg CD27:BV605 (O323), 0.25 µg CD3:SparkBlue550 (SK7), 0.25 µg CD304:AlexFluor647 (12C2), 0.03 µg CD4:BV750 (SK3), 0.5 µg CD45RA:APC/Fire750 (HI100), 0.13 µg CD56:BV711 (5.1.H11), 0.13 µg CD8:BV570 (RPA-T8), 0.5 µg CD95:PE/Cy5 (DX2), 0.13 IgD:PE/Dazzle594 (IA6-2), 0.13 µg PD-1:APC (A17188B, Biolegend), 0.06 µg CD38:PerCP/eFluor710 (HB7, Thermo Fisher), 0.06 µg CD177:PE/Vio770 (REA258), 0.05 µg CD25:PE (REA570, Miltenyi, Bergisch Gladbach, Germany).

Antibodies were incubated for 45 min on ice in the dark. In parallel, in order to analyse the proportions of apoptotic leukocytes, Apotracker Green (Biolegend) was added directly to the blood samples according to the manufacturer’s instruction with an incubation for 30 min on ice in the dark. In order to lyse erythrocytes, 2.2 mL Fixative-Free Lysing Solution (Thermo Fisher) was added and incubated for 20 min at room temperature in the dark. Following this, 0.03 µg 4′,6-diamidino-2-phenylindole was added as a live/dead discriminator and incubated for 5 min. Finally, data acquisition was performed on the Cytek Aurora flow cytometer running on the SpectroFlo Software version 2.2.0.3 (Cytek Biosciences, Fremont, CA, USA). Analysis of flow cytometry data was done using the FlowJo software version 10.7 (FlowJo, Ashland, OR, USA) with the manual gating strategy shown in Appendix A. Dimension reduction of down-sampled and concatenated data sets was performed using the FlowJo plugin for the algorithm “uniform manifold approximation and projection” (UMAP) [15].

### 2.3. Multipathway Posphoprotein Analyses

T lymphocytes were obtained by processing 2 mL anticoagulated whole blood using CD3 MicroBeads on an automated magnetic activated cell sorting device (autoMACS, Miltenyi) according to the manufacturer’s instructions. The fraction enriched for CD3^+^ cells was centrifuged and washed with PBS and frozen at −80 °C. Cells from the CD3^−^ fraction were washed with Running Buffer (RB, Miltenyi) and erythrocytes were lysed by adding 5 mL Fixative-Free Lysing Solution (Thermo Fisher), mixing and incubating for 15 min at room temperature. The lysis was quenched by adding 10 mL RB and mixing. Subsequently, cells were pelleted, suspended in RB and B lymphocytes were isolated by utilizing CD19 MicroBeads (Miltenyi) on an autoMACS. The fraction enriched for B lymphocytes was then washed and the cell pellet was frozen thereafter as described above. For the isolation of monocytes we used 2 mL anticoagulated blood and CD14 MicroBeads (Miltenyi). The CD14^+^ fraction was processed as mentioned above. The CD14^−^ fraction was submitted to the isolation of granulocytes by CD11b MicroBeads (Miltenyi) similar to the enrichment of CD19^+^ cells.

Proteins were obtained from sorted CD3^+^, CD19^+^, CD14^+^ and CD11b^+^ cells by adding 40 µL MILLIPLEX Map Lysis Buffer containing 100-fold diluted Protease Inhibitor Cocktail Set III (Merck, Darmstadt, Germany) and mixing at 4 °C and 700 rpm for at least 15 min. Samples were subsequently centrifuged at 10,000× *g* for 10 min at 4 °C. Supernatants were diluted fourfold and bulk protein concentrations were determined by using the Pierce BCA Protein Assay Kit (Thermo Fisher) following the manufacturer’s guidelines. The absorbance was measured at 562 nm on the Infinite 200 automated plate reader (Tecan, Männedorf, Switzerland). Subsequently, samples were processed using the MILLIPLEX Multi-Pathway Cell Signaling Assay kits (Merck) for the semiquantitative analysis of either phosphorylated or unphosphorylated cAMP response element-binding protein (CREB), extracellular signal-regulated kinases (ERK)1/2, nuclear factor kappa-light-chain-enhancer of activated B cells (NF-κB), c-Jun N-terminal kinase (JNK), p38, p70S6 kinase, signal transducer and activator of transcription (STAT)3 and STAT5 to the manufacturer’s instructions. In brief, 1–25 µg of total protein was used. Samples where protein concentrations were insufficient were excluded from further analysis. For calibration purposes, unstimulated and stimulated cell lyophilisates were reconstituted to a protein concentration of 2 mg/mL. In detail, the kit provided unstimulated HeLa cells, HeLa cells stimulated with tumor necrosis factor (TNF)α/Calyculin A, MCF7 cells stimulated with insulin-like growth factor 1 and A431 cells stimulated with epidermal growth factor. Serial dilutions were performed and processed in parallel to the samples. Data acquisition was performed on the Luminex 100/200 flow cytometer with the acquisition software xPONENT version 3.1.871.0 (Luminex, Austin, TX, USA).

Data analysis was performed by log10-transformation of raw median fluorescence intensity (MFI) values and annotating arbitrary units (AUs) for the respective standard dilution. In order to obtain AUs for samples, we utilized a linear regression model for the calibration standards onto which transformed sample MFI values were fitted. After back-transformation, the data for unphosphorylated proteins were normalized to the amount of whole protein from the BCA assay. AUs for phosphorylated proteins were normalized to AUs from their unphosphorylated counterparts. For heatmap visualization, the data was normalized to a 0–1 range (z value).

### 2.4. SARS-CoV-2 Specific Antibodies

Plasma samples were obtained by centrifugation of anticoagulated blood from healthy donors and COVID-19 patients, respectively. For the semi-quantitative analyses of IgM and IgG specific for the nucleocapsid (N) or the spike protein subunits S1, S2 and receptor binding domain (RBD), we used the respective MILLIPLEX SARS-CoV-2 antigen panel kits (Merck) to the manufacturer’s instructions. In brief, plasma samples were diluted 100-fold in assay buffer and were incubated with antigen-conjugated beads for 2 h at room temperature with shaking at 600 rpm. After washing, PE-conjugated anti-human IgG/IgM detection antibodies were added to the samples followed by incubation for 1.5 h at room temperature and 600 rpm. Samples were then washed, immersed in sheath fluid (Merck) and analyzed on the Luminex 100/200 flow cytometer (Luminex). For the detection of trimeric spike protein specific IgA we used an enzyme-linked immunosorbent assay according to the manufacturer’s specifications (Euroimmun, Lübeck, Germany). Absorbances were detected at 450 nm (A_450_) on the Infinite 200 automated plate reader (Tecan).

### 2.5. Cytokine Analysis

For the determination of cytokine concentrations in plasma samples, the LEGENDplex Human Anti-Virus Response Panel (Biolegend) was used containing capture beads and detection antibodies for Interleukin (IL-)1β, IL-6, IL-8, IL-10, IL-12p70, Interferon (IFN)α, IFNβ, IFNλ1, IL-29, IFNλ2/3, IL-28, IFNγ, TNFα, interferon gamma-induced protein 10/C-X-C motif chemokine ligand 10 (IP10/CXCL10) and granulocyte-macrophage colony-stimulating factor (GM-CSF). The protocol was followed using the manufacturer’s instructions. Data acquisition was performed using the Cytek Aurora (Cytek Biosciences).

### 2.6. Statistics

Data analysis and visualization were performed using R (version 3.5.1), InStat (GraphPad, San Diego, CA, USA) and SPSS (IBM, Armonk, NY, USA). Contingency tables were compared by the Fisher’s exact test. Normality of data sets was evaluated using the Kolmogorov-Smirnov test. Comparisons for multiple groups were done by analysis of variance (ANOVA) under the assumption for normally distributed sample data. Pairwise comparisons were conducted post-hoc with the Tukey-Kramer test. Non-normally distributed data were compared by the Kruskal-Wallis one-way analysis of variance with post-hoc pairwise comparisons using the Mann-Whitney-Wilcoxon (MWW) test and adjustment of *p*-values with the Holm-Bonferroni method. The Spearman rank method was used for correlation analyses. A *p*-value of <0.05 was considered statistically significant.

## 3. Results

### 3.1. Mild-to-Moderate COVID-19 Induces Minor Changes in Immune Cell Subpopulation Distribution and Activation

Twenty-eight participants were recruited from the local COVID-19 test center. Reasons for visiting were either return from high incidence regions, direct contact with SARS-CoV-2 infected individuals or experience of COVID-19 related symptoms, respectively. Eight of these participants tested positive for SARS-CoV-2 but did not require hospitalization due to mild symptoms. Fourteen inpatients were recruited from the division of tropical medicine and infectious diseases. All of them ran non-severe disease courses without the need for intensive care. Among all participants, two patients were available for three consecutive venipunctures, nine for two and eleven for a single one. Table 1 lists the demographic data of patients and healthy donors showing an even distribution of sex and comparable age ranges.

In order to further our understanding of early immune cell responses in the peripheral blood during mild-to-moderate COVID-19, we performed 24-dimensional flow cytometry analyses and in a first approach compared healthy controls with SARS-CoV-2 infected patients from all time points. Our primary goal was to define whether subpopulations of neutrophilic granulocytes, monocytes and lymphocytes of the B or T lineage were specifically enriched during early infection. For that, we performed dimension reductions on our multiparametric data sets by using the embedding algorithm UMAP. Figure 1A shows the topological distributions of immune cell subpopulations based on different surface antigen expression patterns for healthy controls and COVID-19 patients, respectively. While the data imply ample variation for the abundance of CD19^+^ and CD177^+^ populations between both groups, the differences regarding CD3^+^ and CD14^+^ populations were less prominent.

We next sought to investigate whether SARS-CoV-2 infection induced the activation of particular signaling pathways in immune cells. For that, we sorted CD3^+^, CD11b^+^, CD14^+^ and CD19^+^ cells from peripheral blood samples and analyzed from lysates the extent of protein phosphorylation. Figure 1B exemplifies for CD14^+^ monocytes a hierarchical clustering that suggests increases of phosphorylated p38, ERK1/2, JNK, STAT3, STAT5 and CREB in COVID-19 patients. However, when analyzing individual signaling pathways in more detail, variances among groups turned out to be very high and obscured potential differences between patients and controls (Appendix A). Likewise, cluster analyses of phosphorylated signaling molecules in CD19^+^, CD3^+^ or CD11b^+^ cells did not reveal any COVID-19 specific activation of the respective pathways (Figure 1B, Appendix A).

Our broad survey of surface expression- and intracellular activation data therefore hint at rather small-scale alterations among bulk immune cell responses during mild COVID-19 and hence call for a more sophisticated data-dependent stratification.

### 3.2. The Fraction of Plasmablasts in Combination with Antibody Titers Delineate Infection Phases

In order to explain the variation among the COVID-19 samples, we aimed to delineate a time line of early immune events following SARS-CoV-2 infection. However, neither the date of a positive test result for SARS-CoV-2 nor hospital admission are suitable reference points for the classification of different disease phases, as COVID-19 leads to a diverse collection of symptoms that may be delayed or may fail to develop altogether even in the presence of substantial viral loads [16,17,18]. We therefore reviewed our multidimensional flow cytometry data for advanced clues that would allow for the assignment of the various samples to specific disease stages along this timeline and for subsequent comparison of these disease stages with each other and to the controls.

As was recently described for COVID-19, enrichment of a distinct population of the B lineage cells may indicate a shift from the steady state immune constitution and point towards an early adaptive response [19,20,21]. Indeed, we found a transient increase of CD19^+^CD45RA^+^CD27^+^CD38^bright^ plasmablasts in consecutive samples from COVID-19 patients (Figure 2A).

Accordingly, dynamics of early antibody producing cells and class-switching may translate into a time- or disease stage-dependent accumulation of immunoglobulins (Igs) of different isotypes. A very early stage post infection should be devoid of any antibodies and, thereafter, IgM, IgA and IgG specific for viral proteins may appear almost simultaneously. In fact, it has been shown that IgM titers will increase for about two weeks after emergence before they plateau, while amounts of IgA and IgG are expected to increase over a longer period of time [21,22].

Given these assumptions, we analyzed SARS-CoV-2 specific Igs in plasma samples and integrated the dynamics of peripheral IgD^−^ plasmablasts in order to stratify our data set. We thus were able to define four different disease stages: (i) healthy controls that have no antibodies against SARS-CoV-2 and feature a mean of 1.1 ± 0.2% of plasmablasts among CD19^+^CD45RA^+^ lymphocytes, (ii) an incubation phase with still negligible SARS-CoV-2 specific Igs and a mean of 2.9 ± 0.8% of plasmablasts, (iii) early acute infection characterized by elevated IgM, IgG and IgA titers along with a significantly increased proportion of plasmablasts (11.5 ± 2.6%), and (iv) a late acute infection phase defined by a reduction in plasmablasts (4.4 ± 1.1%) together with stable IgM and minor increases in IgA and IgG titres (Figure 2B, Appendix A). Of note, we assumed that age would be a confounding variable influencing some of our data and therefore incorporated analyses of covariants (ANCOVA) in order to eliminate age from the statistical model (Figure 2B and Appendix A). Indeed, we found that age specifically operated on the null hypothesis rejection for the alteration of CD27^+^CD38^dim^ early memory B-lymphocytes which were nonetheless altered between different disease stages when precluding the confounding covariable (Appendix A).

In summary, our data demonstrated that utilizing the dynamics of both plasmablast trajectories and SARS-CoV-2 specific Ig titers allowed for the delineation of patient data into differently progressed disease phases.

### 3.3. Pro-Inflammatory Monocytes and CD16^−^CD177^+^ Granulocytes Follow Different Kinetics during Mild-to-Moderate COVID-19

Building on the fact that early phases of viral infection trigger innate immune responses, we sought to investigate the dynamics of monocyte and granulocyte subpopulations between the above postulated disease phases. Peripheral monocytes were defined via sideward scatter and expression of CD14. The additional analysis of CD16 expression allowed for the differentiation of classical (CD14^+^CD16^−^), anti-viral (CD14^+^CD16^+^) and pro-inflammatory monocytes (CD14^+^CD16^bright^). Figure 3 demonstrates for the contingent of pro-inflammatory monocytes a moderate but statistically not quite significant increase in the incubation phase when compared to healthy controls (*p* = 0.0766). However, upon progression into the acute infection phases, this population of monocytes was significantly reduced (Figure 3). Finally, we did not find any meaningful differences between the remaining monocyte subpopulations and the various phases of the disease (Appendix A).

For granulocytes we gated SSC^hi^ cells and then differentiated subpopulations by their expression patterns for CD11b, CD16 and CD177. While most of the granulocyte subpopulations were unaltered between all disease stages, we found that CD16^−^CD177^+^ cells were slightly increased during the incubation phase and early acute infection. However, during late acute infection, there was a significant increase in this population compared to healthy controls (Figure 4 and Appendix A).

In summary, the definition of early disease phases allowed for the detection of different kinetics among innate immune cells. While the proportion of pro-inflammatory monocytes increased rather early in the response to SARS-CoV2, CD16^−^CD177^+^ granulocytes were found to accumulate in the late acute infection stage.

### 3.4. Incubation and Early Acute Infection Phases Are Characterized by an Increase in Activated Cytotoxic T Cells and IP-10

In order to characterize in detail the cytotoxic T cell response during incubation as well as early and late acute infection phases, CD8^+^ cells were analyzed for their activation (CD25, CD38), exhaustion (PD-1) and differentiation status (CD45RA, CD27, CD127). While the fraction of naïve (CD45RA^+^CD27^+^) CD8-positive T cells was strictly age dependent, there were no further associations with any of the disease phases. However, the incubation phase was characterized by a significant relative increase in CD45RA^+^CD38^+^CD27^−^ cytotoxic T cells (Figure 5A), indicating an activated and terminally differentiated status [23]. This fraction of CD38^+^CD27^−^ cytotoxic T cells gradually declined over the different phases of acute infection (Figure 5A,B). Following a different kinetic, CD45RA^+^CD38^+^CD27^+^ cytotoxic T cells, suggestive of TSCM (stem cell memory) started to increase during incubation yet reached a significant difference compared to healthy controls only during late acute infection (Figure 5A and Appendix A). Likewise, CD45RA^+^CD127^−^CD27^+^ cells, indicative of TEMRA (effector memory re-expressing RA), started to increase during early acute infection and were significantly elevated during late acute infection (Appendix A). Moreover, all disease phases featured an elevated expression of CD38 on CD8^+^ cells (Figure 5C and Appendix A).

Finally, we observed a significantly increased fraction of PD-1^+^ cells among cytotoxic T cells during the early acute infection phase. These PD-1^+^ cells were either CD27^−^ or CD27^+^ and already in the decline during late acute infection (Figure 5D,E and Appendix A). In contrast to the alterations among CD8^+^ subpopulations, we did not observe similarly prominent changes among CD4^+^ T helper cell differentiation/activation stages and the various disease phases (Appendix A). Neither were there significant changes among CD25^+^CD127^+^ regulatory T cells nor NK cells (Appendix A).

Finally, we investigated the kinetics of 13 cytokines in an anti-virus response panel and found that IP-10 (CXCL10) was transiently elevated during both the incubation phase and early acute infection before returning to baseline in late acute infection (Figure 6 and Appendix A). In support of mild-to-moderate COVID-19 courses in our cohort, we did not detect a cytokine storm for any of the analytes measured. We did, however, find slight increases of IL-6, IL-8, IL-12, TNFα, IFNγ, IFNα2, IFNλ1, IFNβ, GM-CSF and IL-10, mostly for the acute infection stages. However, none of these exhibited a meaningful effect size yet were subject to high variance (Appendix A).

In summary, while the increase of activated CD38^+^CD27^−^ and activated or exhausted PD-1^+^ cytotoxic T cells was mostly restricted to the incubation phase and early acute infection, respectively, we found the activation marker CD38 to be overexpressed during all disease phases.

### 3.5. Incubation and Early Acute Infection Are Characterized by T Lymphopenia Unrelated to Apoptosis

In order to also assess quantitative differences among the major leukocyte populations during the early disease phases, we determined absolute numbers of live granulocytes, lymphocytes and monocytes. The picture emerging revealed lymphopenia during incubation and early acute infection which slowly recovered during late acute infection (Appendix A). This lymphopenia was based on a decrease in T lymphocytes and affected both T helper cells and cytotoxic T cells (Figure 7A). Quantitative differences among B cell, granulocyte, and monocyte bulk populations did not reach statistical differences (Appendix A). To rule out apoptosis as the reason for T lymphopenia, we calculated the respective proportions of apoptotic cells among live leukocytes. However, no significant variations emerged that paralleled the fluctuating numbers of live T cells nor were there any other populations affected (Figure 7B and Appendix A). Apoptosis therefore could not be held responsible for the decrease of peripheral T lymphocytes during incubation and early acute infection.

### 3.6. IP-10 in Patients Correlates with Different Immune Parameters Than in Healthy Controls

Finally, we performed correlation analyses for all immune parameters assessed. As for the healthy controls, the only significant correlation existed between the fraction of pro-inflammatory monocytes (CD14^+^CD16^bright^) and serum concentrations of IP-10 (Appendix A). In contrast, in the patients, serum IP-10 correlated significantly with CD38^+^CD27^−^ terminally differentiated cytotoxic T cells (Appendix A). Moreover, patient serum concentrations of all anti-S isotypes correlated significantly with each other and also with the fraction of immature neutrophilic granulocytes.

## 4. Discussion

By analyzing longitudinal trajectories of immune responses to SARS-CoV-2 in patients experiencing mild-to-moderate disease courses, we were able to define three different stages (Figure 8). The first one comprises the incubation phase, which for the patients manifested with symptoms that led to voluntary testing or—in the case of elderly or multimorbid patients—to hospitalization. The immune response in this phase of disease progression is characterized by significant increases in pro-inflammatory monocytes (CD14^+^CD16^bright^), activated cytotoxic T cells (CD38^+^), and serum IP-10 paralleled by a significant decrease of peripheral T helper cell numbers. This incubation phase is followed by the early acute infection where peripheral plasmablasts (CD27^+^CD38^bright^) are transiently yet significantly elevated, as are exhausted cytotoxic T cells (PD-1^+^). IP-10 is still significantly upregulated, while both helper and cytotoxic T cells are significantly reduced. Virus-specific IgM, IgG and IgA antibodies are detectable, and except for IgM, still on the rise. Finally, during late acute infection, most values appear to be returning to normal except for virus-specific antibodies and immature neutrophils which are now significantly elevated compared to the healthy controls (Figure 8).

We here show that the transient increase in peripheral plasmablasts in combination with seroconversion and relative increases in Ig subclasses serve as reference points to time early phases of COVID-19. While plasmablasts increase between days one and nine after onset of symptoms and peak around day nine [22], SARS-CoV-2 specific antibodies can be detected as early as three to six days post onset of symptoms [20,21]. Importantly, Ig subclasses in the serum did not appear to follow the classical IgM–IgG–IgA sequence of events, as shown e.g., for the seroconversion following HIV infection [24,25]. Instead, whether IgM or IgG appeared first seemed to depend on the antigen with anti-N and anti-S2 inducing a robust IgG and anti-S1 and anti-RBD inducing an IgM response first. IgA appeared simultaneously with either IgM or IgG; however, while IgM on average plateaued during early acute infection, IgG and IgA were still on the rise during late acute infection. We here did not specifically assess SARS-CoV-2 neutralizing antibodies yet refer to a previous publication demonstrating that IgA dominates the early neutralizing response [22].

Taking the alteration of peripheral plasmablasts and virus-specific antibodies as a basis to define various phases of COVID-19 allowed for the detection of small scale alterations among bulk immune cell responses which otherwise were obscured. For example, we here show that the incubation phase is characterized by a sharp increase in pro-inflammatory monocytes and, indeed, this increase has previously been confirmed for both acute Dengue and HI virus infections [26,27]. During Dengue virus infections, CD14^+^CD16^bright^ pro-inflammatory monocytes were suggested to take on an important role, as they were shown to migrate to the draining lymph nodes where they encounter and activate antigen-specific memory B cells to differentiate into plasmablasts that secrete IgG and IgM [26]. Moreover, correlations with serum concentrations of IP-10 hinted at pro-inflammatory monocytes as the main producers of this chemokine [28]. We can confirm this correlation for the healthy controls only. However, in the patients, IP-10 correlated significantly with terminally differentiated cytotoxic T cells. We therefore postulate that IP-10, which is held responsible for the cytokine storm in critically ill patients, is produced by cytotoxic T cells [29,30,31].

In our study, immature neutrophils, characterized by a lack of CD16 expression, continuously increased over time, yet reached a significant difference compared to healthy controls only during late acute infection. Interestingly, a handful of longitudinal studies on hospitalized COVID-19 patients have shown that severity was associated with an increase in neutrophil numbers and that these numbers were persistently elevated, even at clinical recovery [32]. Even though we here did not assess severe cases, our findings also suggest an unresolved inflammation at a time when most immune parameters had returned to normal.

What was conspicuously absent in our study was any major alteration among CD4^+^ T helper cell subpopulations, including T_regs_. In contrast, there were significant changes among cytotoxic subpopulations, among them an elevated CD38 expression, suggesting activation over all disease phases analyzed. Moreover, we observed a significant increase in terminally differentiated (CD27^−^) cytotoxic T cells during incubation, which gradually declined during later phases. In chronic HIV infection, this population was shown to be inefficient and to correlate with viral burden [33,34]. Clearly, its role in COVID-19 awaits further elucidation. An opposing trend—increase during acute infection—was observed for both the CD45RA^+^CD38^+^CD27^+^ and CD45RA^+^CD127^−^CD27^+^ populations, reminiscent of gradual differentiation into stem cell memory (TSCM) and effector memory (TEMRA), respectively [23,35]. Finally, there was also a significantly increased fraction of PD-1^+^ cytotoxic T cells during early acute infection. Even though PD-1 expression on SARS-CoV-2 specific CD8^+^ cells was previously associated with an activated phenotype [36], prolonged antigen stimulation and sustained expression of inhibitory receptors like PD-1 can ultimately lead to impaired T cell functionality and exhaustion [37].

As for significant alterations among the bulk populations of B cells, T cells, NK cells, and monocytes, during the early phases of COVID-19, we only observed T-lymphopenia, and this was more pronounced for the helper than the cytotoxic T cells. However, this drop in peripheral T cells was not paralleled by an increase in apoptotic cells, suggesting that, rather than being eliminated, T cells possibly relocated to the sites of infection or to lymphatic tissues. Of note, we did not observe any significant alteration among NK cells, neither among the bulk nor among subpopulations. Likewise, there were no alterations among monocyte/NK or monocyte/granulocyte ratios, which confirms previous findings about mild-to-moderate COVID-19 [32,38].

A major limitation of this study is the small sample size of COVID-19 patients. Additionally, not all patients were available for all consecutive venipunctures. We therefore propose that future studies should consider a study design in which shorter intervals for sample acquisitions are incorporated for a larger cohort. Furthermore, we did not document the participants’ underlying health status and are therefore unable to assess whether co-morbidities affected the variance within and between groups. Finally, in contrast to what is shown in this study, delineation of acute disease phases on the basis of SARS-CoV-2 specific antibody dynamics only applies to primary infection, as convalescent patients would already have developed a humoral response prior to re-infection.

In summary, the present study defines a plasmablast-antibody landscape that defines different disease stages during mild-to-moderate COVID-19. These disease stages were further characterized by means of cellular parameters and identify the time frame where cytokinogenesis was most prominent. Upcoming research needs to focus on comparable disease stages during severe COVID-19 in order to allow for the identification of landmarks where resolution of viral infection segregates from progressing immune dysregulation and fatal outcomes. Moreover, as global efforts are underway to identify and develop new antiviral and immunomodulatory therapeutics that reduce COVID-19 related hospitalization and deaths, the need arises for the right timing [39]. Indeed, early type I IFN treatment seemed to be critical for therapeutic success [40]. However, prophylactic immunomodulation might not be feasible for most COVID-19 patients due to high costs and potential adverse events. Therefore, identifying immunological landmarks will enhance a therapeutic potential towards benefits rather than harm.

## Figures and Tables

**Figure 1 viruses-14-00203-f001:**
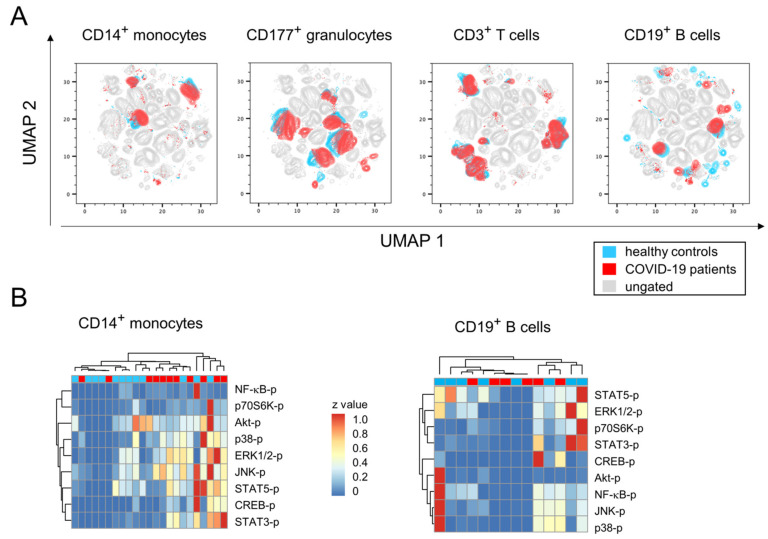
Multidimensional flow cytometry and phosphoprotein analyses revealed minor changes of bulk immune cell response during mild-to-moderate COVID-19. (**A**) UMAP projection of flow cytometry data show the topological distribution of immune cell populations based on differentially expressed surface antigen patterns between healthy controls and COVID-19 patients. (**B**) Heatmaps of normalized protein phosphorylation data from sorted CD14+ monocytes (left) and CD19+ B lymphocytes (right) demonstrated contingent hierarchical clustering of healthy controls and COVID-19 patients.

**Figure 2 viruses-14-00203-f002:**
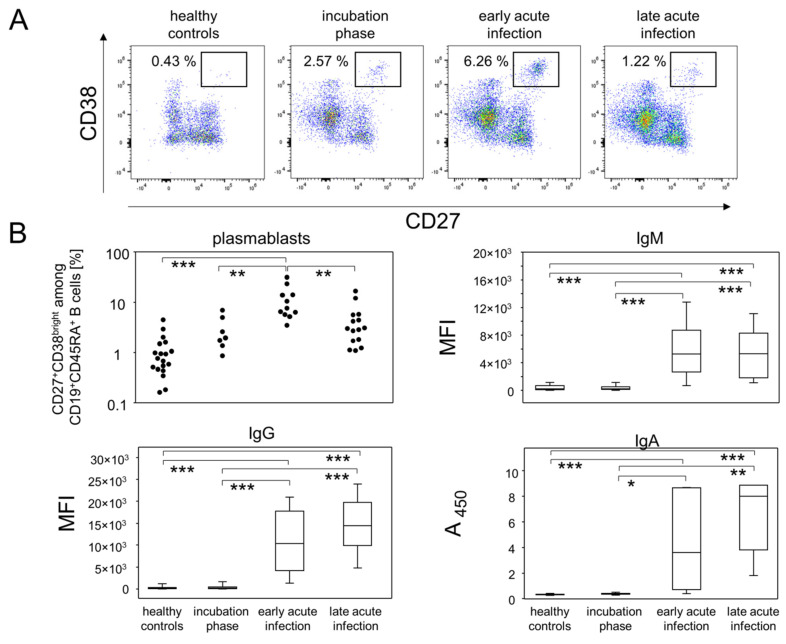
Kinetics of plasmablast proportions and SARS-CoV-2 specific immunoglobulin enrichment delineated disease stages during mild-to-moderate COVID-19. (**A**) Representative pseudocolor plots of CD38 and CD27 expression data of CD19^+^CD45RA^+^ B lymphocytes showed the disease stage-dependent increase of plasmablast proportions (rectangular gate). (**B**) Top left: Quantitative data of plasmablast abundances at different disease stage. Top right, bottom left and bottom right: Semi-quantitative data of SARS-CoV-2 specific IgM, IgG and IgA titers in plasma samples. The data for anti-RBD, -S1, -S2 and –N were combined for IgM and IgG, respectively.* *p* < 0.05; ** *p* < 0.01; *** *p* < 0.001; Tukey-Kramer test for plasmablasts, Kruskal-Wallis test with Dunn’s correction for multiple comparisons for Antibody isotypes. MFI: Median fluorescence intensity. A_450_: Absorbance at 450 nm.

**Figure 3 viruses-14-00203-f003:**
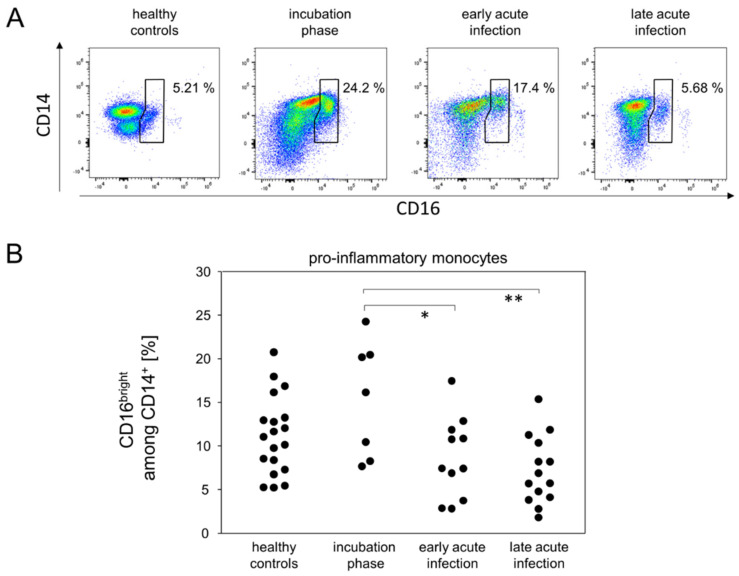
Pro-inflammatory monocytes were more abundant during the incubation phase. (**A**) Representative pseudocolor plots of CD14 and CD16 expression data of SSC^med^ monocytes demonstrated kinetics for the enrichment of pro-inflammatory monocytes (polygonal gate). (**B**) Quantitative data of pro-inflammatory monocyte proportions. * *p* < 0.05; ** *p* < 0.01; Tukey-Kramer test.

**Figure 4 viruses-14-00203-f004:**
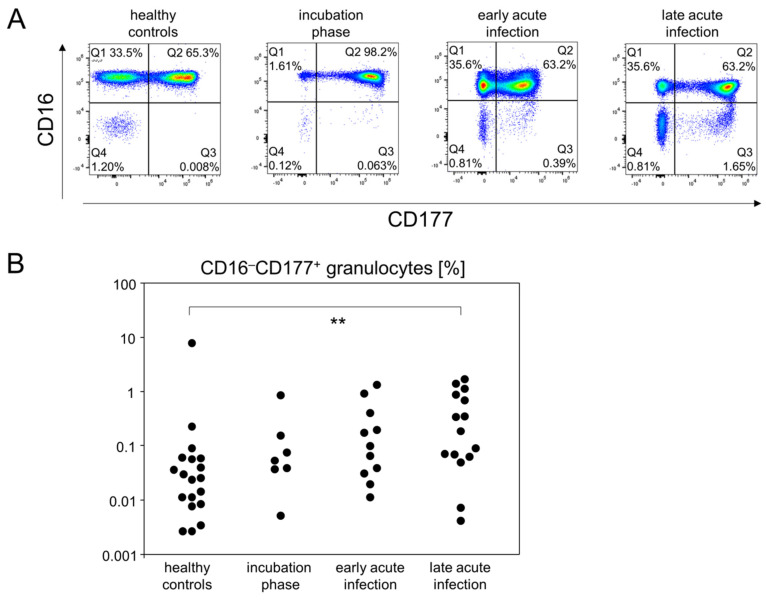
The portion of CD16^−^CD177^+^ granulocytes gradually increased until the late acute infection phase. (**A**) Representative pseudocolor plots of CD16 and CD177 expression data of SSC^hi^ leukocytes showed that CD16^−^CD177^+^ granulocytes specifically emerged at the late acute infection phase (Q3 gate). (**B**) Quantitative flow cytometry data indicated gradually increasing abundances of CD16^−^CD177^+^ granulocytes. ** *p* < 0.01; Tukey-Kramer test.

**Figure 5 viruses-14-00203-f005:**
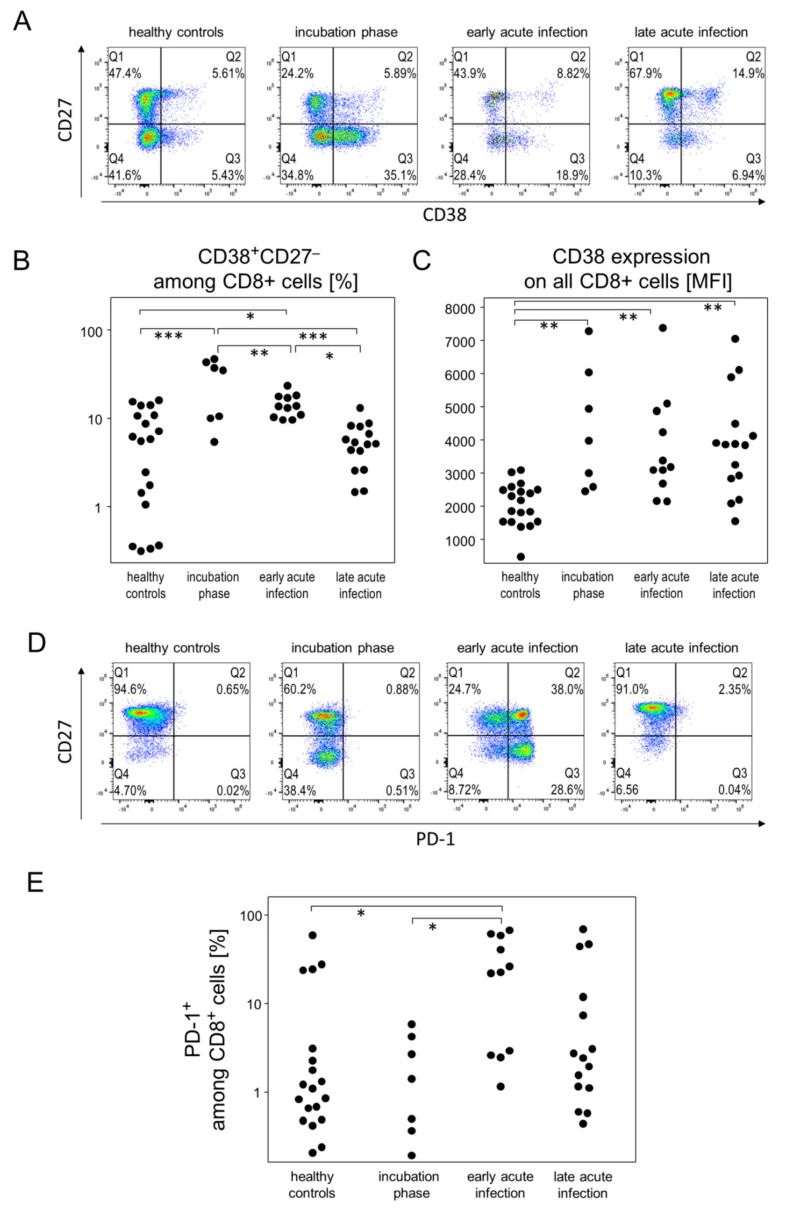
The scope of cytotoxic T lymphocytes activation depended on the respective disease stage during mild-to-moderate COVID-19. (**A**) Representative pseudocolor plots of CD38 and CD27 expression data of CD8^+^ T cells demonstrated the accumulation of the CD38^+^CD27^−^ subpopulation (Q3 gate) during the incubation and early acute infection phases. (**B**) Quantitative flow cytometry data of CD8^+^CD38^+^CD27^−^ T cell proportions. (**C**) CD38 expression by CD8^+^ T cells at the different disease stages. (**D**) Representative plots of PD-1 and CD27 expression data showed an increase of PD-1^+^ T cells that are either CD27^+^ (Q2 gate) or CD27^−^ (Q3 gate). (**E**) Quantitative data of CD8^+^PD-1^+^ T cell proportions showed their enrichment during the incubation and acute infection phases. * *p* < 0.05; ** *p* < 0.01; *** *p* < 0.001; Tukey-Kramer test.

**Figure 6 viruses-14-00203-f006:**
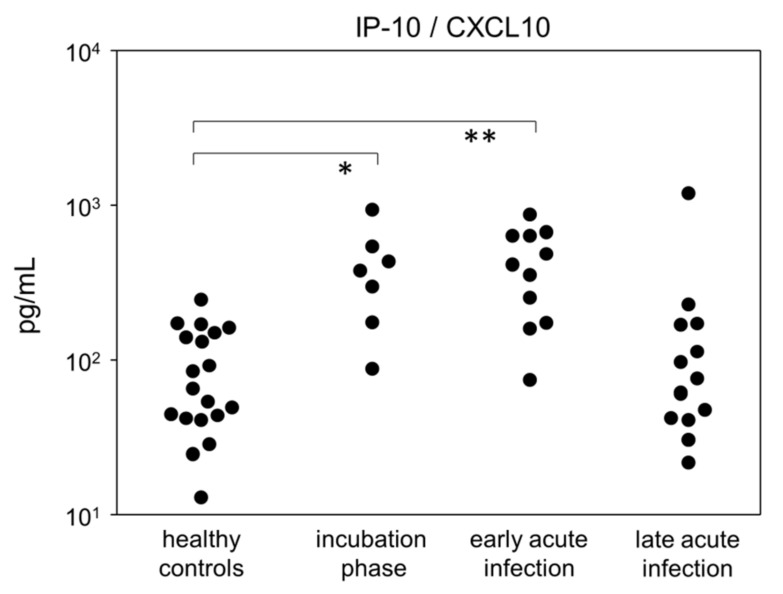
Quantitative data of IP-10 concentrations in plasma samples at different disease stages. * *p* < 0.05; ** *p* < 0.01; Tukey-Kramer test.

**Figure 7 viruses-14-00203-f007:**
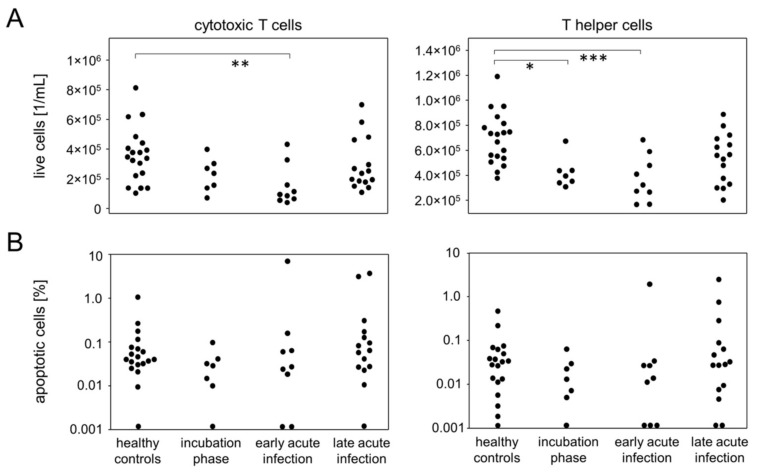
Mild-to-moderate COVID-19 induced apoptosis-independent CD4^+^ and CD8^+^ T lymphopenia at the incubation and early infection phases. (**A**) Total cell numbers of cytotoxic T cells (left) and T helper cells (right). (**B**) Quantitative data of apoptotic cell proportions among cytotoxic T cells (left) and T helper cells (right). * *p* < 0.05; ** *p* < 0.01; *** *p* < 0.001; Tukey-Kramer test.

**Figure 8 viruses-14-00203-f008:**
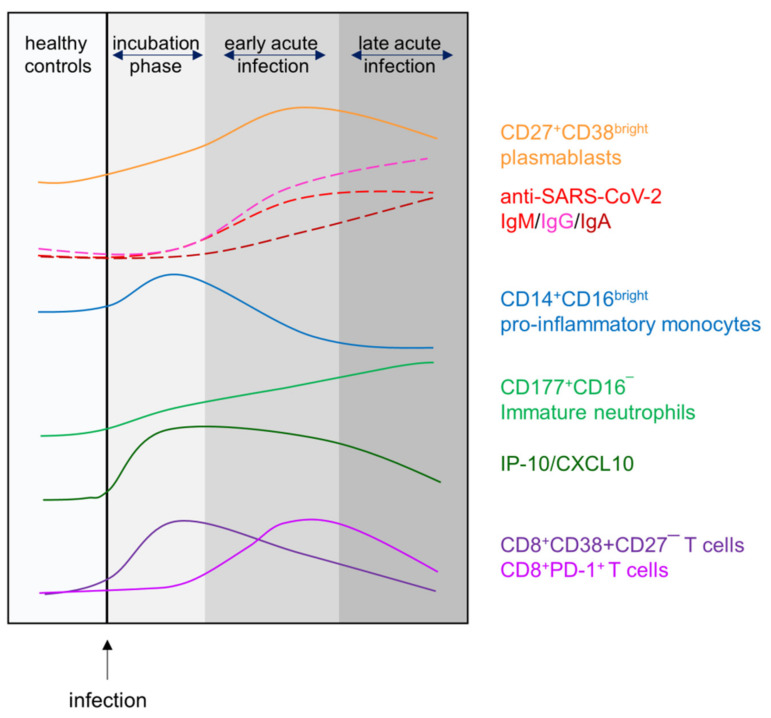
Differential kinetics of immune cell populations along distinct disease stages during mild-to-moderate COVID-19.

**Table 1 viruses-14-00203-t001:** Demographics of patients and controls.

	Controls (*n* = 19)	COVID-19 Patients (*n* = 22)	*p*-Value
sex (male/female)	9/10	11/11	1 *
median age (min–max)	68 (22–89)	57 (22–85)	0.574 ^#^

* resulting from Fisher’s exact test, ^#^ resulting from Mann-Whitney Wilcoxon test.

## Data Availability

All data generated or analyzed during this study are included in this published article (and its Appendix A).

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
