# Peer review of "Distinguishing Incubation and Acute Disease Stages of Mild-to-Moderate COVID-19"

_viruses, 2022, doi:10.3390/v14020203_

Round 1
Reviewer 1 Report
The original manuscript ID: Viruses-156416 on “Distinguishing Incubation and Acute disease Stages of Mild-to-Moderate Covid-19 by Michael Muller et al., from Dr. Brigitte Müller-Hilke’s lab.
The manuscript described methods to decline the stage for mild to moderate sars-cov-2 patients infection based on cellular and humoral and immunity. The methods and strategies are interesting and conclusions match the results. However, limited cases in this report, were analyzed and this study only suite to primary Sars-CoV-2 infection, especially the antibody response (IgM, IgG, IgA) in the Covid19 patients.
Table 1 indicate 22 sample were collected patients however, data summarized here only 7 for incubation phase, and earlier infection: 11 sample and late infection 15 sample. Thus, total number of samples are limited.
Page 4 line 158-161, Nucleoprotein, Spike S1 and Spike S2 antibody detection were used in this study. Which antibodies showed in the Figure 2, are they Spike specific or nucleocapsid specific?
If authors would present antibodies against nucleocapsid and spike respectively in Figure 8, would show more interesting results, due to nucleocapsid antibodies appear much earlier than spike antibodies.
The integrating of the kinetics of plasma blasts and SARS-CoV-2 specific antibody to distinguish three stages: incubation, early acute and late acute phase. However, this classification might not be suitable for all chronic patients such as myeloma and leukemia diseases. To whose patients have very high plasma cells and T or B cells. Therefore, authors would be necessarily to point out: in this study, sample collection was excluded from patients with other chronic diseases.
Minor questions:
Abstract: line 18, Within 7 days; In material and methods: Line 75: venipuncture on days 2 and 6. Which is correct?
Figure 2. B: the incubation phase and late acute phase versus healthy control might be significant too for CD27+CD38bright/plasma blasts.
Figure 3. B: CD16 Bright might reduce in the late acute phase versus healthy control. Statistic significant?
Author Response
Response to the reviewers’ comments:
Reviewer #1
The reviewer made thoughtful and meaningful suggestions on how to clarify some aspects of our study. As follows, we provide a point-by-point list of revisions that refer to these suggestions.
Comment 1: “The manuscript described methods to decline the stage for mild to moderate sars-cov-2 patients infection based on cellular and humoral and immunity. The methods and strategies are interesting and conclusions match the results. However, limited cases in this report, were analyzed and this study only suite to primary Sars-CoV-2 infection, especially the antibody response (IgM, IgG, IgA) in the Covid19 patients.”
Yes, it is correct that our results and the conclusions we have drawn apply to primary acute SARS-CoV-2 infection, only. Accordingly, we updated the discussion section in order to clarify this limitation (Page 16): “[…], in contrast to what is shown in this study, delineation of acute disease phases on the basis of SARS-CoV-2 specific antibody dynamics only applies to primary infection, as convalescent patients would already have developed a humoral response prior to re-infection.”
Comment 2: “Table 1 indicate 22 sample were collected patients however, data summarized here only 7 for incubation phase, and earlier infection: 11 sample and late infection 15 sample. Thus, total number of samples are limited.”
We agree that the small samples sizes are a major limitation of this study. We therefore added a paragraph that refers to this and other limitations within the discussion (Page 15): “A major limitation of this study is the small samples size of COVID-19 patients. Furthermore, not all patients were available for all consecutive venipunctures. We therefore propose that future studies should consider a study design in which shorter intervals for sample acquisitions are incorporated for a larger cohort.” Please note, that the revision based on your first comment can also be found in this paragraph.
Comment 3: “Page 4 line 158-161, Nucleoprotein, Spike S1 and Spike S2 antibody detection were used in this study. Which antibodies showed in the Figure 2, are they Spike specific or nucleocapsid specific?”
We apologize for not clarifying this in the legend for Figure 2. In fact, we combined the results of anti-RBD, S1, -S2 and –N for IgM and IgG, respectively. The figure legend was updated accordingly. Please see the attached file in which you can find the separated data for each antigen. The data show that antibody dynamics were quite similar for the different antigens.
Comment 4: “If authors would present antibodies against nucleocapsid and spike respectively in Figure 8, would show more interesting results, due to nucleocapsid antibodies appear much earlier than spike antibodies.”
As shown in the attached file, we did not observe anti-N immunoglobulins emerging earlier than antibodies that are specific to the other SARS-CoV-2 antigens. We therefore do not believe that adding another line into the graph would show more.
Comment 5: “The integrating of the kinetics of plasma blasts and SARS-CoV-2 specific antibody to distinguish three stages: incubation, early acute and late acute phase. However, this classification might not be suitable for all chronic patients such as myeloma and leukemia diseases. To whose patients have very high plasma cells and T or B cells. Therefore, authors would be necessarily to point out: in this study, sample collection was excluded from patients with other chronic diseases.”
As already stated in the methods section (2.1, page 2), we did not document the underlying health status of the participants. Although we consider it unlikely that several severe co-morbidities like myeloma or leukemia have accumulated in our small cohort, we cannot exclude their possible influence on within-group variance. Hence, we expanded the limitation paragraph within the discussion section as follows (Pages 15 & 16): “Furthermore, we did not document the participants’ underlying health status and are therefore unable to assess whether co-morbidities impacted on the variance within and between groups.”
Comment 6: “Abstract: line 18, Within 7 days; In material and methods: Line 75: venipuncture on days 2 and 6. Which is correct?”
We apologize if our phrasing was not perfectly clear. However, both statements are correct depending on the method for counting. When you start counting with day 0, seven days are reached upon day 6. Hence, the method section was improved and now states: “After an initial withdrawal of peripheral blood on the day of testing or admittance to the hospital (day 1), subsequent venipunctures followed on days 3 and 7.”
Comment 7: “Figure 2. B: the incubation phase and late acute phase versus healthy control might be significant too for CD27+CD38bright/plasma blasts.”
We agree that the mean plasmablast proportion at the late acute infection phase seems elevated when compared to healthy controls. However, according to our analysis, this increase was not statistically significant due to high variance and type I error correction for multiple comparisons (p = 0.178, Tukey-HSD).
Comment 8: “Figure 3. B: CD16 Bright might reduce in the late acute phase versus healthy control. Statistic significant?”
The apparent difference between healthy controls and late acute infection was short of reaching statistical significance (p = 0.086, Tukey-HSD).

Reviewer 2 Report
The current article by Michael Muller and al., reviews the current literature on the distinguishing incubation and acute disease stage of mild-to-moderate COVID-19.The title of the paper is in line with the body of the manuscript. The topic is current and timely and very important for the world's scientific community in the current period of the global COVID-19 pandemic. The authors have written a clear and detailed review and the material is well presented, although the authors do not express particular personal ideas on the future prospects for investigation. The references used are suitable and it is new and updated material, I believe that the article can be accept.
Author Response
Response to the reviewers’ comments:
Reviewer #2
We thank the reviewer for his positive feedback. Please note that some revisions were made to the manuscript in response to the other reviewer’s comments and suggestions.